Accurate spatiotemporal predictions of daily stream temperature from statistical models accounting for interactions between climate and landscape

Siegel Jared E. jared.siegel@noaa.gov derajlegeis@gmail.com 1 2
Volk Carol J. 2
1 Ocean Associates, under contract to Northwest Fisheries Science Center, National Oceanic and Atmospheric Administration , Seattle , WA , United States of America
2 South Fork Research, Inc , North Bend , WA , United States of America
Burns Douglas
Electronic publication date: 2019 Nov 12
Publication date: 2019
Volume: 7
Electronic Location ID: e7892
Received 2019 Apr 1; Accepted 2019 Sep 13
Copyright: ©2019 Siegel and Volk
Copyright year: 2019
Copyright holder: Siegel and Volk
License: This is an open access article distributed under the terms of the Creative Commons Attribution License, which permits unrestricted use, distribution, reproduction and adaptation in any medium and for any purpose provided that it is properly attributed. For attribution, the original author(s), title, publication source (PeerJ) and either DOI or URL of the article must be cited.
License URL: https://creativecommons.org/licenses/by/4.0/

Keywords: Stream temperature, Climate change, Statistical model, Spatiotemporal, Interaction, Autocorrelation, Salmon, GAM

Funding: South Fork Research, Inc The Bonneville Power Administration (BPA Project number 2003-017-00) Funding was provided by South Fork Research, Inc. and the Bonneville Power Administration (BPA Project number 2003-017-00). The funders had no role in study design, data collection and analysis, decision to publish, or preparation of the manuscript.

==============================
Spatial and temporal patterns in stream temperature are primary factors determining species composition, diversity and productivity in stream ecosystems. The availability of spatially and temporally continuous estimates of stream temperature would improve the ability of biologists to fully explore the effects of stream temperature on biota. Most statistical stream temperature modeling techniques are limited in their ability to account for the influence of variables changing across spatial and temporal gradients. We identified and described important interactions between climate and spatial variables that approximate mechanistic controls on spatiotemporal patterns in stream temperature. With identified relationships we formed models to generate reach-scale basin-wide spatially and temporally continuous predictions of daily mean stream temperature in four Columbia River tributaries watersheds of the Pacific Northwest, USA. Models were validated with a testing dataset composed of completely distinct sites and measurements from different years. While some patterns in residuals remained, testing dataset predictions of selected models demonstrated high accuracy and precision (averaged RMSE for each watershed ranged from 0.85–1.54 °C) and was only 17% higher on average than training dataset prediction error. Aggregating daily predictions to monthly predictions of mean stream temperature reduced prediction error by an average of 23%. The accuracy of predictions was largely consistent across diverse climate years, demonstrating the ability of the models to capture the influences of interannual climatic variability and extend predictions to timeframes with limited temperature logger data. Results suggest that the inclusion of a range of interactions between spatial and climatic variables can approximate dynamic mechanistic controls on stream temperatures.

Introduction

Spatial and temporal patterns in stream temperature are primary factors determining species composition, diversity, and productivity in stream ecosystems (Vannote et al., 1980). Ichthyofauna and most other stream dwelling species are ectotherms, thus temperature dictates the rates of metabolic and physiological processes influencing growth, development, and the timing of life history events. Stream temperature also impacts biota by affecting the dissolved oxygen content of water and biogeochemical processes such as nutrient cycling, decomposition rates, and eutrophication. Many human activities, such as deforestation (Holtby, 1988) and flow regulation by dams and diversions (Sinokrot & Gulliver, 2014), have major effects on stream temperature. Additionally, climate change has already caused increases in stream temperature (Isaak et al., 2010; Isaak et al., 2012; Kaushal et al., 2010; Ruesch et al., 2012) and changes in the timing and magnitude of discharge (Luce & Holden, 2009; Riedel & Larrabee, 2016), with effects on the productivity and spatial distributions of resident species (Lawrence et al., 2014). Due to the ecological importance of stream temperature combined with the potential impact of human activities, there is substantial interest among water and fisheries managers in improving our understanding of stream temperature patterns and influences.

Investigations into the effects of stream temperature on biota are best supported by temperature metrics at spatial and temporal scales relatable to the biological responses of interest. For example, stream temperature affects numerous life history events of Pacific salmon; including the timing and success of the spawning migration (Crozier, Scheuerell & Zabel, 2011), pre-spawn mortality rates (Bowerman et al., 2017), egg incubation times and larval survival (Pankhurst & Munday, 2011), the size and timing of emergence (Beacham & Murray, 1990), juvenile energetic demands and growth rates (Crozier et al., 2010), and the size and timing of smolting (Sykes & Shrimpton, 2010). Each of these life history events occurs at temporally distinct time periods and in spatially distinct habitats. Accordingly, a comprehensive examination of the effects of stream temperatures on Pacific salmon would benefit from spatially and temporally continuous estimates of stream temperature within the watershed they inhabit, allowing for a full exploration of temperature effects and the summarization of temperature predictions into specific metrics of interest.

Most stream temperature data are collected by loggers which capture instantaneous point measurements at pre-determined temporal intervals. Modeling techniques are then utilized to expand the utility of logger data by making hindcast/forecast predictions across space or time and to describe influences on stream temperature from climate/spatial variables (e.g., McNyset, Volk & Jordan, 2015; Segura et al., 2015; Letcher et al., 2016; Turschwell et al., 2016; Isaak et al., 2017a; Isaak et al., 2017b). Water temperature models are generally classified as either deterministic or stochastic/statistical (Caissie, 2006; Benyahya et al., 2007). Deterministic models are based on mathematical representation of the underlying physics of heat exchange between the river and the surrounding environment (e.g., Caissie, Satish & El-Jabi, 2007). In contrast, most statistical models utilize air temperature (e.g., Mohseni, Stefan & Erickson, 1998; Pilgrim, Fang & Stefan, 1998) as a surrogate for net heat exchange processes in the absence of detailed information on heat fluxes, as both are dependent on solar radiation (Webb et al., 2008). Consequently, statistical models require less site-specific data and are generally easier to scale than deterministic models (Benyahya et al., 2007).

To produce accurate predictions across temporal (diel, daily, seasonal, and annual) and spatial (microhabitat, reach, tributary, and watershed) gradients, a statistical model needs to parameterize relationships that approximate the complex interactions between geography and climate that determine stream temperature. Climatic factors, such as air temperature, snowpack melt, and stream discharge levels affect heat exchange within streams, driving seasonal and interannual variability in temperature (Caissie, 2006). The spatial characteristics of streams leads to heterogeneity in the localized responses to climate within and across watersheds. For example, higher gradient watersheds (Mayer, 2012) and more shaded stream reaches (Holtby, 1988; Johnson, 2004) tend to be cooler in the summer. Higher elevation watersheds as well as groundwater and snowmelt dependent streams are less sensitive to air temperature changes (Luce et al., 2014; Lisi et al., 2015; Isaak et al., 2016; Winfree et al., 2018). Similarly, hyporheic exchange (Arrigoni et al., 2008) and tributary confluences (Ebersole, Liss & Frissell, 2003) can lead to localized zones with distinct temperature patterns. The resulting complexity in spatiotemporal patterns in stream temperature can lead to streams with diverse thermal profiles in similar climatic regions (Fullerton et al., 2015).

However, most statistical stream temperature modeling techniques are limited in their ability to account for the influence of variables changing across spatial and temporal gradients, which restricts the universality of derived relationships. Consequently, many statistical modeling approaches reduce the spatial or temporal component of monitoring data, greatly limiting their utility. For example, while the majority of statistical models depend on the air temperature/stream temperature relationship, this relationship varies spatially within basins (Pilgrim, Fang & Stefan, 1998; Mayer, 2012; Steel, Sowder & Peterson, 2016) and across basins (Arismendi et al., 2013; Luce et al., 2014; Mauger et al., 2017; Winfree et al., 2018) as a consequence of distinct landcover and geomorphological influences. Accordingly, air temperature driven models are often fit to single sites as the validity of derived relationships may be spatially limited. The relationship between air and water temperature is not consistent over years (Arismendi et al., 2014) and seasons (Lisi et al., 2015) due to climatic effects, such as variability in discharge levels and the influence of snowpack. While a few recent statistical modeling studies have attempted to account for variation in the nature of the relationship with air temperature in diverse ways (e.g., Li et al., 2014; Segura et al., 2015; Jackson et al., 2018), modeling efforts could benefit from further exploration and discussion of effective methodologies.

Spatial modeling techniques, which utilize statistical auto-correlation to describe how upstream sites influence downstream sites, have recently become popular to predict stream temperature across entire watersheds (e.g., Peterson et al., 2013; Isaak et al., 2014; Jackson et al., 2017). Most spatial modeling techniques require dense monitoring networks (Marsha et al., 2018) and, with a few exceptions (e.g., Jackson et al., 2018; Hocking, Neil & Letcher, 2018), have primarily been used to predict temporally summarized metrics, such as the mean August max weekly temperature (Isaak et al., 2017a; Isaak et al., 2017b) or the maximum weekly mean stream temperature (Ruesch et al., 2012), as opposed to continuous estimates of temperature. This is likely a consequence of spatial correlations and covariate relationships changing with climatic variability (Steel, Sowder & Peterson, 2016; Jackson et al., 2018). For example, stream temperatures are likely to be more correlated across space during periods of high flows due to shorter water residency times and increased thermal inertia limiting the effect of air temperature and other influences. Consequently, models that depend on such autocorrelation techniques or other hierarchical methods may see large declines in precision when predicting into new temporal or physical spaces which are uninformed by such methods (e.g., Hocking, Neil & Letcher, 2018).

Despite clear theoretical evidence for the influence of covariates on stream temperature being conditional on other climatic and spatial influences, few statistical modeling studies have attempted to extensively utilize interactions to approximate such physical mechanism (e.g., Hocking, Neil & Letcher, 2018; Jackson et al., 2018), and the utility of many interactions remains largely unexplored. Our objective was to identify and describe important interactions between climate and spatial variables and to utilize these relationships to model reach-scale spatially and temporally continuous patterns in daily mean stream temperature across complex watersheds. The justification for this investigation is based on three assertions: (1) the shifting influence of the effects of variables with climate and space can be approximated by including interaction terms based on mechanistic principles, (2) accounting for this variability in stream temperature with variable relationships, as opposed to statistical methods, can inform mechanisms and allowing for the expansion of predictions across space and time, and (3) few studies have extensively explored the utility of utilizing interactions to model stream temperature and thus there is potential for substantial improvement.

To achieve our objective, we compared two statistical modeling techniques, simple linear models and general additive models (GAMs), fit to daily mean temperature data throughout the entire year using a suite of climatic and spatial variables. GAM models can easily account for non-linearities in relationships, such as between stream temperature and air temperature, with non-parametric smoothers (Wood, 2006). However, the benefits of their flexibility are balanced by their tendency to overfit data. Separate models of both types were generated for four Columbia River tributary basins in the Pacific Northwest (United States) containing federally listed populations of Chinook salmon (Oncorhynchus tshawytscha) and steelhead (Oncorhynchus mykiss), which have thermally sensitive life histories. We explored whether spatiotemporal variability could be accounted for using only variables and interactions rather than utilizing hierarchical or autocorrelation statistical modeling techniques. Modeled relationships were validated by predicting testing datasets composed of years and logger sites completely distinct from the model training datasets.

Materials & Methods

Study watersheds

We modeled stream temperature in four tributary watersheds of the Columbia River located in the northwestern United States, the Wenatchee River, the Chiwawa River, the Middle Fork John Day River (M.F. John Day), and the Tucannon River (Fig. 1). These watersheds were selected to represent a range in size (474 to 3,452 km2), wetness (mean annual discharge of 5 to 91 m3/s) and climatic regimes for exploring relationships among physical and climatic variables. All four study watersheds are influenced by winter snowpack and seasonal climate patterns, leading to high spring flows during the snowmelt and low flows during the late summer/early fall dry period.

Figure 1 Map of study basins and environmental data collection sites.

Maps of study watersheds showing the stream logger sites used for model training and model testing (validation) datasets for the Wenatchee/Chiwawa (A), the M.F. John Day (B), and the Tucannon (C). The location of stream gages and climatic stations where data for environmental covariates were collected are also shown. Due to limitations in the spatial distribution of site coverage in the M.F. John Day and the Tucannon, the modeled stream networks were restricted to the upper basin and mainstem channel respectively.

Wenatchee River. The Wenatchee River drains the east side of the Cascade Mountains (Washington, USA) and flows southwest to the Columbia River. It is the largest of the study watersheds, with an area of 3,452 km2 and mean discharge of 91 m3/s. The upper reaches are characterized by high alpine mountains and national forest lands while the lower reaches support significant agricultural production (e.g., apples and cherries). Small high elevation cirque glaciers influence some of the tributaries and are most prominent in the White River. Lake Wenatchee (10.0 km2), located at the confluence of the White and Little Wenatchee tributaries, is a prominent feature in the central watershed.

Chiwawa River: The Chiwawa River is a tributary of the upper Wenatchee River, with its confluence just below Lake Wenatchee. We included analysis of the Chiwawa River separately as a test of how scale affects model accuracy within the same geographical region. The river drains high-mountain and forest lands, representing about 14% of the drainage area of the Wenatchee watershed and 16% of the mean annual discharge (area 474 km2, mean discharge 14 m3/s).

Middle Fork John Day River: The M.F. John Day is a tributary of the upper John Day draining the dry pine forests of the Blue Mountains in eastern Oregon. With an area of 2,088 km2, the basin is about 60% the size of the Wenatchee River but is significantly drier with very low summer flows (mean discharge 7.28 m3/s). The watershed contains habitats ranging from near-alpine to sagebrush steppe in the lower reaches near its confluence with the North Fork John Day River. Due to low coverage of sites in the lower basin, we only modeled the watershed at the confluence with Slide Creek and above (Fig. 1, modeled area 1,178 km2).

Tucannon River: The Tucannon is a tributary of the Snake River draining the Blue Mountains in the southwest corner of Washington State. The upper basin is characterized by mountainous pine forests, while the lower watershed contains dry sage bush and agricultural lands. The Tucannon is a relatively dry system and is similar to the M.F. John Day in both discharge and size (area 1,300 km2, mean discharge 4.81 m3/s). With relatively few temperature monitoring locations on tributaries, we restricted the analysis to the mainstem Tucannon from the confluence with the Snake to the headwaters in the Blue Mountains.

Stream temperature data

For each watershed, all available stream temperature data from two regional databases were downloaded and utilized for model training and testing respectively (Fig. 2). Stream temperature data sourced from the Columbia Habitat Monitoring Program (ISEMP/CHaMP, 2017) was downloaded for all study watersheds for years 2012–2017 and used as the model training datasets. Additional stream temperature data for the Wenatchee training dataset was provided by the Washington Department of Fish and Wildlife (Jeremy Cram, personal communication). Model testing data for these watersheds, collected from loggers at distinct sites and during different years, was downloaded from the NorWest Regional Stream Temperature Database (Isaak et al., 2017a; Isaak et al., 2017b, https://www.fs.fed.us/rm/boise/AWAE/projects/NorWeST.html). Testing data was collected on the Wenatchee and Chiwawa from 2003–2011,1997–2011 for the M.F. John Day, and 2001–2011 for the Tucannon. Both testing and training datasets contained years with diverse climate conditions (Figs. S1–S3). Although potential distinctions in collection procedures between the datasets could cause discrepancies, combining datasets gave us the longest possible time series to examine the ability of models to account for interannual variability. While training and testing datasets were controlled for quality by their respective publishers before downloading, we visually examined all data for anomalies before utilization in this study leading to the removal of a minimal amount of data. A small number of sites (3 in the M.F. John Day and 1 in the Tucannon), which demonstrated highly restricted seasonal fluctuations in stream temperature compared to directly adjacent sites, were removed from consideration as these sites were deemed likely to be directly located in groundwater springs representing micro-habitats, as opposed to mixed mainstem waters. Instantaneous measures of stream temperature data were summarized to daily mean temperatures. Training datasets were largely distributed evenly across the year while testing datasets were disproportionately represented by summer data, particularly in the M.F. John Day and the Tucannon (Table 1).

Figure 2 Model fit and validation years for each study basin.

Representation of study methodology: Seasonal models (spring and fall) were fit across all years of data in training datasets (2012–2017). Models were subsequently used to predict daily data in the testing datasets (1997–2011) in the Wenatchee, Chiwawa, M.F. John Day, and Tucannon river basins.

Table 1 Dataset characteristics table.

Characteristics of model fitting and testing datasets shown for entire datasets and by month. The average temporal representation of sites varied across rivers and datasets. Effective coverage is the average percent of days with logger coverage for all sites within the respective time series (data days/[sites*years*365]). Data days by month is from all sites across all years. While data in fitting datasets were relatively evenly distributed across the year, data in validation datasets disproportionately cover summer months, particularly in the M.F. John Day and the Tucannon.

 	Training datasets	 	Testing datasets	
Basin	Wenatchee	Chiwawa	M.F.J.D.	Tucannon	 	Wenatchee	Chiwawa	M.F.J.D.	Tucannon	
year range	2012–2017	2012–2017	2012–2017	2012–2017	 	2003–2011	2003–2011	1997–2011	2001–2011	
sites	40	11	45	41	 	36	3	179	21	
sites per year (avg.)	28	8	22	34	 	20	3	34	13	
effective coverage (%)	52%	56%	33%	67%	 	42%	63%	16%	55%	
data days	45,629	13,479	32,407	59,953	 	32,929	4,171	63,561	25,072	
data days per year (avg.)	7,605	2,247	5,401	9,992	 	3,659	463	4,889	2,279	
data days by month										
Jan.	4,051	1,209	2,493	5,052	 	1,862	232	434	514	
Feb.	3,585	1,080	2,237	4,609	 	1,686	202	392	477	
Mar.	4,115	1,209	2,506	5,028	 	1,972	235	365	527	
Apr.	4,031	1,170	2,473	4,803	 	1,978	237	713	531	
May	4,158	1,219	2,587	4,962	 	2,103	238	2,325	2,886	
Jun.	3,988	1,135	2,777	4,757	 	2,294	242	8,115	3,590	
Jul.	3,640	1,109	3,170	4,997	 	3,779	513	14,367	4,128	
Aug.	3,747	1,176	3,369	5,517	 	5,544	806	15,078	4,278	
Sep.	3,214	937	3,021	5,376	 	4,928	729	13,364	3,973	
Oct.	3,866	1,100	2,907	5,361	 	3,020	349	5,897	3,250	
Nov.	3,575	1,050	2,412	4,681	 	1,901	190	1,986	462	
Dec.	3,659	1,085	2,455	4,810	 	1,862	198	525	456	

Spatial and climatic covariates

A suite of spatial and climatic covariates was considered for stream temperature models (Table 2). Spatial variables were estimated continuously across the basin by stream reach but contained no temporal component. In contrast, climatic variables were temporally continuous (daily values with the exception of Snow April 1st, which was an annual value) but were collected from point source environmental monitoring stations.

Table 2 Table showing the variables considered for stream temperature models.

Temporal (A) and spatial (B) variables considered as covariates in model selection with description of variable calculation, spatial and temporal characteristics, and rationale for inclusion.

Variable	Description	Spatial	Temporal	Rationale	
A. Temporal variables	
D	Day of year (days)	NA	1 day means	Accounts for seasonal changes in length of days and solar angle	
T3a	Average air temperature from 3 day period before predicted day (°C)	From one point at headwaters	5 day means	The influence of air temperature on stream temperature accumulates over time	
T5a	Average air temperature from 5 day period before predicted day (°C)	From one point at headwaters	3 day means	The influence of air temperature on stream temperature accumulates over time	
TΔa	Difference between utilized averaged temperature variable (T3a/T5a) and mean temperature the day of predictions (°C)	From one point at headwaters	1 day means	Air temperature effects temperature in real time	
S	Snowpack depth	From one point at headwaters	1 day means	More snowpack contributes colder water to streamflow	
SA1	April 1st snowpack depth (cm)	From one point at headwaters	1 day means	Magnitude of late snowpack has prolonged effect on stream temperature into the summer (delayed discharge, riparian growth)	
F	Flow at USGS gage (m3/s)	From one point near mouth	1 day means	Higher discharge creates more insulation against atmospheric influences. Seasonally different relationship (cooling in summer, warming in winter)	
B. Spatial variables	
E	Average elevation of catchment area (m)	Summarized by catchment area of the stream reach	NA	Catchments with higher terrain will have cooler streams even if the site is at a lower elevation	
EΔ	Difference between E and the site specific elevation (m)	Summarized by catchment area of the stream reach	NA	Higher elevation sites experience cooler air temperatures and are closer to cooler headwaters	
A	Catchment area of site (km2)	Summarized by catchment area of the stream reach	NA	Sites further from headwaters have more time to be effected by atmospheric temperatures	
BFI	Estimated Base flow index (mean low flow ÷ mean annual discharge)	Summarized by catchment area of the stream reach	NA	Areas with higher groundwater influence will have mitigated stream temperatures (lower highs and higher lows). Developed by Wolock (2003)	
L	Percentage of catchment covered by lakes	Summarized by catchment area of the stream reach	NA	Lakes slow down water leading to increased atmospheric warming	
SL	Slope of stream reach	Summarized by stream reach	NA	Steeper streams move cooler water downstream faster	
FC/FR	Forest cover percentage of reach contributing area (FR) or catchment (FC)	Summarized by reach contributing area and catchment area of the stream reach	NA	Forested areas provide more stream shading and retention of moisture/snowpack	

All climatic variables were collected from environmental monitoring stations located within watersheds (Fig. 1). Air temperature and snowpack data were collected from the environmental monitoring stations with the highest elevation and most continuous time series in each basin as provided by the National Oceanic and Atmospheric Administration’s National Centers for Environmental Information (NCEI, 2018; https://gis.ncdc.noaa.gov/maps/ncei/summaries/daily). All flow data (F) was collected from the most downstream USGS stream gage in each watershed (United States Geological Survey, 2017; https://waterdata.usgs.gov/nwis/rt). Gaps in coverage of climate variables were filled with a linear spline, though missing data was minimal. Since the effect of air temperature can be delayed due to thermal inertia (Letcher et al., 2016), air temperature was summarized as a time-lagged variable. The primary air temperature variables considered in models represented the average temperature during the three or five day period before the day of interest (T5a and T3a). To limit multicollinearity, information for air temperature on the day of interest was included as the difference between T5a or T3a and the mean temperature on the day of interest (TΔa). Accordingly, TΔa represents if it got colder (negative value) or warmer (positive value) on the day of interest in comparison to prior days. While values of climatic variables in both datasets exhibited substantial interannual variability, variability was largely similar between the two datasets in all cases (e.g., Figs. S1–S3).

All spatial variables were estimated for stream reaches segmented at confluences within the 2–6th stream order of the 1:24,000 National Hydrography Dataset (NHD, McKay, Bondelid & Dewald, 2012) using ArcGIS version 10.3 (ESRI, 2011). Long reaches were further segmented to a maximum of 3 km. The majority of spatial variables were summarized by reach specific upstream catchment areas. This creates spatially smoothed predictions as reaches close together have similar catchment areas. To account for the effect of elevation on stream temperature we utilized two variables, the average elevation of the catchment area (E) and the difference between E and the reach specific elevation (EΔ). Utilizing EΔ instead of the reach specific elevation reduced multicollinearity. Other spatial variables included the catchment area (A), the Base Flow Index (BFI) as described in the National Hydrography Plus Dataset, the proportion of catchments covered by lakes (L), the reach specific slope (SL), and the percent forest cover of the reach contributing area and entire catchment respectively (FR/FC).

As with most other statistical models for stream temperatures, models depended heavily on the relationship between stream temperature and air temperature. To account for how this relationship changes across space and time, we considered interactions between air temperature (T5a or T3a) and all other variables (spatial and climatic). Interactions between all other variables and day of year (D) were also considered to capture smoother seasonal shifts in relationships that may be resilient to daily fluctuations in air temperature. Variables were allowed to interact with an air temperature variable or D, but not both. As higher elevation sites are more directly influenced by seasonal snowpack than low elevation sites, interactions between daily values of snow depth (S) and the average elevation of the catchment (E) were considered. Finally, interactions between the spatial variables of catchment area (A) and catchment elevation (E) were considered to help distinguish between high and low elevation tributary streams, which are likely to have distinct discharge and temperature regimes.

Model fitting and selection

Two statistical modeling methods, simple linear and general additive models (GAMs, Wood, 2006), were used to fit models to stream temperature data from the training datasets (years 2012–2017) for each study watershed (Table 1, Fig. 2). To help account for hysteresis in stream temperature, or distinct air/water temperature relationships during the spring warming period due to the influence of snowpack (Harvey et al., 2011, Lisi et al., 2015), separate models were fit to the spring warming period and the fall cooling. While GAM models have the flexibility to account for non-linearities in relationships and thus could have been fit to cover the entire year, seasonal models were utilized to allow for comparisons with linear models. The date of the maximum stream temperature in each watershed as predicted by a smoother fit to all data in the training datasets and the first day of the new year were used to split data into distinct spring and fall training datasets (Fig. S4). Models were fit to all available data in the respective spring and fall training datasets.

When fitting GAM models, the smoothness, or “wiggliness” of the modeled relationships is controlled by the number of splines, or knots. We manually limited the number of knots as automated techniques, such as penalized regression splines, were found to consistently produce over-fit relationships that did not align with hypothesized effects (e.g., Fig. S5). All GAM model variables were limited to 3 knots, with the exception of averaged air temperature variables and day of year variables, which were allowed 5/4 knots to account for substantial non-linearity in the relationship air temperature and stream temperature (Mohseni, Stefan & Erickson, 1998). Explorations found that when more flexibility was given to modeled relationships the resulting models had higher testing dataset prediction error, even if model fits were improved. Interactions in GAM models were fit utilizing the “by” function in R package mgcv, which parameterizes a linear interaction to a smoothed term: f1(x1)* x2 (Wood, 2018). We also explored utilizing two way smoothed interactions and tensor product interactions, but found that these options tended to overfit the data (see Wood, 2006 for more details).

For model selection we performed a backwards stepwise variable selection procedure. We started by fitting global models including all considered variables and interactions (as listed in Table 3). We found that information criterion, such as AIC or BIC, were not adequate for model selection. Due to the large size of our datasets, these modeling techniques tended to support the inclusion of variables that did not align with hypothesized effects and consequently produced poor testing dataset predictions. Additionally, while our variables contain substantial spatial and temporal information, it is likely that they do not fully account for all spatial and temporal autocorrelation, which may lead to inflated variable significance values (Isaak et al., 2014). To overcome this challenge, climatic, spatial, and interaction terms were retained in linear regression and GAM models only if: (1) their estimated effect aligned with the hypothesized effect of the variable; and (2) the variable was a significant contributor to model performance (P < 0.05). Individual variables and interactions that did not meet this criteria were removed one at a time and the model was reassessed at each step until a final model was selected. Variable effects were visualized using the R package visreg (Breheny & Burchett, 2017) and compared to hypothesized effects as described briefly in Table 2 and in more depth in the Supplementary Document. Model predictions below zero were changed to zero to represent freezing.

Table 3 Table showing the variables and interactions utilized for each model and fitting/testing statistics.

Model selection table showing retained covariate (Cov.) and interactions (Int.) in GAM and linear models for the Wenatchee, Chiwawa, M.F. John Day and Tucannon watersheds (A) and fitting and model prediction statistics for selected models (B). Retained variables are represented by grey boxes. Glaciers and lakes are minimally present or not present in the M.F. John Day and the Tucannon watersheds and thus were not considered as a covariates. Knots (K) were limited in GAM model fitting to reduce risk of overfitting. Selected model AIC values, as well as model fit and testing dataset prediction metrics, are provided including the root mean squared error (RMSE) and Nash-Stutclife Coeficient (NSC). Statistics are shown for the model fit (Train), cross validation (C.V.), and testing dataset predictions (Test).

Cov.	Int.	K	Wenatchee	Chiwawa	M.F.J.D	Tucannon	
			Spring	Fall	Spring	Fall	Spring	Fall	Spring	Fall	
		 	GAM	Lin.	GAM	Lin.	GAM	Lin.	GAM	Lin.	GAM	Lin.	GAM	Lin.	GAM	Lin.	GAM	Lin.	
A. Variable selection	
Climate	
D	T3/5	5																	
T3a	 	5	 	 			 	 			 	 			 	 			
T5a	 	4			 	 			 	 			 	 			 	 	
TΔa	 	3																	
F	T3/5	3											 				 	 	
S	T3/5	3	 	 			 	 			 				 	 			
SA1	D	3											 				 		
Spatial	
A	E	3																	
Spatial*Climate	
A	T3/5	3			 		 	 	 	 	 	 							
A	F	3					 		 	 			 	 			 	 	
E	S	3									 	 			 	 			
E	D	3																	
EΔ	T3/5	3										 						 	
BFI	T3/5	3	 		 	 	 	 			 		 	 	 	 	 	 	
L	D	3									NA	NA	NA	NA	NA	NA	NA	NA	
FC	T3/5	3	 	 	 	 	 		 	 		 	 	 	 	 	 	 	
FR	T3/5	3			 	 	 	 	 	 	 	 	 	 	 	 	 	 	
SL	T3/5	3							 	 					 	 	 	 	
G	T3/5	3	 	 	 	 	 	 	 	 	NA	NA	NA	NA	NA	NA	NA	NA	
B. Selected model statistics	
Terms	 	 	14	27	12	24	12	22	11	22	13	21	12	19	11	15	8	13	
ΔAIC	 	 	0	1426	0	954	0	334	0	297	0	4256	0	884	0	4391	0	3337	
RMSE Train	 	0.95	1.12	0.97	1.15	0.62	0.63	0.78	0.81	1.11	1.42	1.27	1.46	0.85	0.91	1.08	1.15	
RMSE C.V.	 	1.09	1.06	1.35	1.35	0.67	1.17	0.84	1.10	1.30	1.44	1.54	1.53	0.87	0.92	1.09	1.16	
RMSE Test	 	1.18	1.30	1.32	1.44	0.70	0.78	0.93	0.89	1.52	1.81	1.56	1.80	0.88	1.01	1.00	1.01	
 	 	 	 	 	 	 	 	 	 	 	 	 	 	 	 	 	 	 	
NSC Train	 	0.96	0.95	0.95	0.95	0.98	0.97	0.95	0.96	0.96	0.95	0.94	0.94	0.97	0.96	0.95	0.95	
NSC Test	 	0.94	0.93	0.93	0.91	0.98	0.97	0.95	0.96	0.90	0.85	0.89	0.85	0.97	0.96	0.95	0.95	

While creating continuous daily predictions provides flexibility in how models could be utilized, many uses will likely involve summarizing daily estimates to specific periods of interest (e.g., mean temperatures during the spawning migration). To explore the effect of temporally aggregating daily predictions on model accuracy we measured the change in prediction accuracy when aggregating daily predictions to monthly predictions of mean stream temperature. All model fitting and subsequent analyses were performed using the statistical computing program R version 3.4.1 (R Core Team, 2014).

Model validation

Each model was used to predict all available data in the respective training and testing datasets (Fig. 2). To validate the accuracy of model predictions we calculated the root mean squared error (RMSE) where yi is the observed value for the ith observation and y ˆi is the predicted value; RMSE= ∑i=1ny ˆi−yi2n.

We calculated three distinct RMSE statistics for each model: (1) For the model fit to the training dataset (RMSE Train), (2) utilizing a leave one site out cross-validation procedure (RMSE C.V.) in which the entire multi-year time series of daily mean stream temperature values from individual sites were left out of the training dataset, then predicted using the resulting models, and (3) for predictions of the testing datasets which was composed of distinct sites and years from the training datasets (RMSE Test). Since we contend that the RMSE Test. represents the truest test of the ability of models to account for spatiotemporal dynamics in stream temperature, we focus most of the discussion on this result.

The Nash-Sutcliffe model efficiency coefficient (NSC) was also used to estimate the goodness of fit of predictions, where y¯ is the mean stream temperature in the respective testing datasets; NSC=1−∑i=1ny ˆi−yi2 ∑i=1nyi−y¯2.

A NSC value of 1 represents a perfect fit and a negative value suggests the model is worse than using the mean.

Sensitivity analyses

Sensitivity analyses of model testing dataset predictions were performed on the selected GAM and linear models, varying both the number of sites and the number of years used in model fits independently. For site sensitivity, selected models with all sites were compared against models fit utilizing data from 10–35 sites at intervals of five, with the exception of the Chiwawa which only contained eleven sites total in the training dataset. For each site count, 100 iterations of randomly chosen site combinations were taken out of the all available sites. Each model was re-fit utilizing only training data from the subset of sites for each iteration and the resulting model was utilized to re-predict the entire testing dataset. The mean RMSE Test for all iterations for each site count was estimated. A sensitivity analysis for the number of years required was similarly performed varying the number of years utilized from 3–6.

Results

Selected climate and spatial variables for estimating stream temperature

In this text we provide a brief explanation of the variables and interactions found to be useful in describing spatiotemporal patterns in stream temperature in the study watersheds (Table 3). A more thorough discussion of these interactions is provided in the Supplementary Document. Covariate relationships with stream temperature that were consistent in importance and form across watersheds are described as “universal” while those that were watershed or model specific are labeled “local”.

Climate: Relationships with climatic variables were generally universal in all study basins and largely the same suite of climatic variables and interactions were used across watersheds (Table 3). Air temperature (T5a, T3a, and TΔa), snowpack (S, SA1), flow (F), and day of year (D) variables were all influential in describing spatiotemporal patterns of stream temperature. For air temperature, T5a was the preferred variable in spring models while T3a was selected for fall models. This is likely a consequence of generally lower levels of discharge in the fall, and thus less thermal inertia, leading to a more rapid influence of climatic conditions. For snowpack variables, S interacting with air temperature (T5a or T3a) was generally utilized in fall models, while SA1 interacting with D was utilized in spring models. In fall models, high daily values of S were found to mitigate the effect of air temperature (cooler stream temperatures when it is hot, warmer when it is cold). For spring models SA1 interacting with D helped account for a small cooling effect later into the summer that occurs when snowpack is deep. While we would expect the extent of snowpack to be more influential during the spring warming period due to seasonal snowpack melting, much of this effect was accounted for in the relationship with F interacting with T5a (Fig. 3). Similar to snow depth in the fall models, higher flows were found to relate to mitigated extremes in stream temperature in all spring models and in the fall Wenatchee and Chiwawa models. F was not found to be influential in the M.F. John Day and Tucannon fall period models, potentially due to mitigating groundwater influences in these dry basins buffering interannual variability in temperature during the low-flow season. The estimated depression in stream temperatures with high discharge during the spring period was stronger than in the fall period for the Wenatchee and Chiwawa, likely as a consequence of this relationship also accounting for the impact of the spring snowmelt in these mountainous watersheds.

Figure 3 Example of interaction effect on stream temperature (flow and air temperature).

Conditional surface plots showing the modeled effects of averaged air temperature variables (T5a and T3a) interacting with flow (F) on stream temperature (Tw) for the best spring (A–D) and fall models (E–F) from each of the study watersheds. This effect was not retained in the M.F. John Day and the Tucannon fall models. Relationships are presented holding all other variables in models at median values.

Spatial: In contrast to climatic factors, many spatial relationships were found to have localized importance and smaller networks, the Chiwawa and Tucannon, required fewer spatial covariates. However, catchment area (A) and elevation variables (E & E Δ) had universal importance in all watersheds. Due to flow data coming from a point source, A interacting with E helped distinguished high elevation tributaries from lower elevation tributaries, which are likely to become warmer and drier earlier in the season due to lower snowpack influence. E interacting with D described a smoothed seasonal effect of elevation, which was found to have a more substantial cooling effect in the summer in comparison to the winter. A interacting with F was retained in most models and helped distinguished the effect of flow on tributaries of different sizes. Higher values of Base Flow Index (BFI), slope (SL), lakes (L), and forest cover (FC/FR), were associated with cooler stream temperatures at high air temperatures but were only locally important and thus not included in all watershed models. L was not utilized in the M.F. John Day or the Tucannon. However, in the Wenatchee, which contains Lake Wenatchee in the central watershed and thus has a higher proportion of lake area than other watersheds, a strong lake effect was found leading to warmer stream temperatures during summer months.

GAM and linear model performance

Both GAM and linear Models described spatiotemporal patterns in daily mean stream temperature with high levels of accuracy (RMSE generally ∼1 °C) and goodness of fit (NSC generally above 0.9: Table 3 & Fig. 4). Model performance varied amongst watersheds with the Chiwawa and Tucannon models being the most accurate, followed by the Wenatchee models, and the M.F. John Day models having the highest error. Spring models were generally more accurate than fall models as described by all utilized RMSE metrics (average difference ∼15%). The error of the model cross validation tests were similar but slightly higher than the model fits on average, with RMSE Train, ranging from 0.62 to 1.46 °C (spring GAM Chiwawa model to fall linear M.F. John Day model, respectively) and leave one site out cross-validation RMSE C.V., ranging from 0.67 °C to 1.54 °C for the same models. When utilizing the models to predict the testing datasets composed of distinct sites and data from different years, prediction error (RMSE Test) was only slightly higher on average compared to RMSE C.V. (∼3%) and RMSE Train (∼17%), ranging from 1.81 °C (fall linear M.F John Day model) to 0.70 °C (spring GAM Chiwawa model), and some models actually performed better. RMSE Test was largely consistent across years in the testing datasets which contained substantial interannual variability in climate (1997–2003 to 2011; Fig. 5). Additionally, aggregating daily predictions to monthly predictions markedly increased goodness of fit and reduced RMSE Test in all models by an average of 23% (Fig. 4).

Figure 4 Model validation prediction accuracy results.

Graphs showing the accuracy of the best validation predictions (linear or GAM) for combined fall and spring models versus measured daily (A–D) and monthly (E–H) measurements of stream temperature. Testing dataset prediction statistics are shown for each watershed including the root mean squared error (RMSE Test), the mean absolute error (MAE), and the Nash-Sutcliffe Coefficient (NSC Test).

Figure 5 Prediction accuracy by year.

Graphs showing daily testing dataset model predictions versus measured stream temperature by year in the Wenatchee (A–I), Chiwawa (J–R), M.F. John Day (S–JJ) and Tucannon (HH–RR) watersheds. The root mean squared error for the testing datasets predictions by year (RMSE Test) are also shown.

While there was substantially more variability in model performance across watersheds than across modeling techniques, GAM models were generally more accurate than linear models when all utilized RMSE metrics were compared (average difference ∼12%). The lone exceptions were the RMSE C.V. values for the spring Wenatchee and fall M.F. John Day models and the RMSE Test value spring Chiwawa linear models (Table 3).

While winter months were generally predicted with comparable precision as described by RMSE Test, they also expressed less variability than warmer months. Consequently, it was actually the summer months in which models described a higher proportion of temperature variability as defined by NSC values (Table 4). In isolated cases during late-fall and winter months models were actually less useful than taking the monthly mean as demonstrated by NSC values below 0. When isolating for site-specific predictions of monthly stream temperature, and thus removing all seasonal and spatial patterns, models were generally able to distinguish between colder and warmer years, particularly in summer months exhibiting higher inter-annual variability in stream temperature (e.g., Fig. 6).

While the accuracy and precision of dataset predictions were largely consistent across years and models accounted for much of the spatial variability across the landscape (e.g., Figs. S6 and S7), some spatial and temporal patterns remained in the model prediction error. For example, when visualizing the prediction error over time for selected Wenatchee sites from the training dataset (Site WC503432-000155, Peshastin Creek) and the testing dataset (Site # 219, Nason Creek), we saw clear seasonal patterns in residuals that repeat across predicted years (Fig. 7). These two sites were chosen for presentation because they had near median prediction error (RMSE) compared to all sites in the Wenatchee models. Consequently, they are fairly representative of the form and magnitude of residual patterns seen at other sites. In the case of the Peshastin Creek site in the training dataset, the selected GAM models tend to under-predict temperatures in the spring through early fall (Fig. 7A). In contrast, testing dataset predictions at the Nason Creek site tended to be too low in the summer (Fig. 7B). However, as is the case for most sites, predicted temperatures largely remain a degree or two degrees from measured temperatures throughout the entire multi-year time series.

Sensitivity analyses

While selected GAM models produced better predictions than linear models, accuracy of GAM models degraded more quickly when the number of years and sites was reduced (Table 5). We believe GAM models degraded more quickly due to their flexibility combined with no re-examination of the theoretical basis of the fitted relationships in the sensitivity analysis due to automation of the process. We therefore focus our discussion on the sensitivity results for linear models as this is likely a better representation of the amount of data required to achieve desired model accuracy. Linear model sensitivity to the number of sites depended on the watershed, but no sensitivity analysis for the number of sites was done for the Chiwawa since this watershed only contained 11 sites. Increases in RMSE Test of over 10% occurred in the Wenatchee models when sites were reduced to 20, in the M.F. John Day models when sites were removed down to 15, and in the Tucannon models when sites were reduced down to 10. For the sensitivity test for the number of years, increases only surpassed 10% when reducing each tested time series from 6 years to 3 years. Note that sensitivity analyses were likely conservative since the vast majority of sites were not operational for the entire duration of the fitting time-series (Table 5), and if temporal coverage of fitting sites was higher, we would expect lower site requirements.

Table 4 Monthly prediction statistics table.

Testing dataset prediction error (RMSE) and goodness of fit measures (NSC) for selected GAM and linear models shown for aggregated monthly predictions for the Wenatchee, Chiwawa, M.F. John Day and Tucannon watersheds respectively.

 	Wenatchee	Chiwawa	M.F.J.D.	Tucannon	
 	RMSE Test	NSC Test	RMSE Test	NSC Test	RMSE Test	NSC Test	RMSE Test	NSC Test	
Month	GAM	Lin	GAM	Lin	GAM	Lin	GAM	Lin	GAM	Lin	GAM	Lin	GAM	Lin	GAM	Lin	
January	0.81	1.01	0.35	0.00	0.15	0.21	0.75	0.52	0.58	0.71	−0.09	−0.66	0.51	0.69	0.63	0.32	
February	0.58	0.69	0.50	0.29	0.35	0.34	0.42	0.47	0.58	0.90	−0.11	−1.65	0.46	0.51	0.78	0.72	
March	0.68	0.65	0.55	0.58	0.57	0.31	−0.47	0.57	0.79	0.92	0.19	−0.09	0.57	0.68	0.75	0.65	
April	0.77	0.80	0.66	0.64	0.45	0.38	0.00	0.29	1.19	1.31	0.50	0.40	0.90	0.89	0.80	0.80	
May	0.89	0.94	0.76	0.73	0.25	0.28	0.94	0.93	1.31	1.46	0.61	0.52	0.75	0.74	0.91	0.92	
June	1.16	1.20	0.80	0.78	0.58	0.72	0.94	0.90	1.28	1.56	0.78	0.68	0.60	0.64	0.96	0.96	
July	1.28	1.46	0.78	0.71	0.70	0.91	0.90	0.83	1.36	1.73	0.74	0.57	0.68	0.70	0.96	0.96	
August	1.40	1.47	0.69	0.66	0.87	0.78	0.74	0.79	1.38	1.71	0.64	0.44	0.73	0.72	0.95	0.96	
September	0.90	1.00	0.77	0.72	0.46	0.41	0.76	0.81	1.09	1.45	0.70	0.47	0.60	0.58	0.95	0.96	
October	0.82	0.92	0.77	0.71	0.44	0.43	0.86	0.87	1.15	1.31	0.63	0.52	0.77	0.78	0.88	0.87	
November	0.99	1.37	0.78	0.57	0.87	0.70	−6.09	−3.52	1.28	1.46	0.08	−0.19	0.71	0.95	0.85	0.74	
December	1.00	1.19	0.59	0.42	1.02	0.38	−8.62	−0.32	1.50	1.59	−3.02	−3.56	0.91	1.13	−0.33	−1.03	
Year	1.03	1.15	0.95	0.94	0.62	0.60	0.98	0.98	1.26	1.56	0.92	0.88	0.70	0.72	0.98	0.98	

Discussion

Interactions and modeling performance

Climate interacts with spatial heterogeneity on the landscape to produce complex spatiotemporal patterns in stream temperature across watersheds. Our results suggest that much of the variability inherent in this complex process can be approximated in statistical models by utilizing simple interactions terms between spatial and climatic variables. Dependent on these interactions, both GAM and simple linear models predicted spatiotemporal patterns in daily mean values of stream temperature across all four of the complex mountainous study watersheds with a level of accuracy and precision which compares well to other modeling studies with similar objectives (McNyset, Volk & Jordan, 2015; Turschwell et al., 2016; Jackson et al., 2017). Additionally, the ability of models to predict stream temperature at distinct sites and years compared to what was utilized in fitting provides increased confidence that models capture interactions that drive seasonal and interannual variability in stream temperature, as opposed to unique patterns present within a dataset. This result suggests that by accounting for interactions, models are not only capable of providing quality estimates of stream temperature throughout the spatial and temporal extent of a monitoring network, but could be utilized to fill temporal and spatial monitoring gaps and to extend time series when logger data is limited.

Figure 6 The ability of models to distinguish between years by month.

Monthly aggregated testing dataset model predictions from selected GAM models shown versus measured temperatures by month for the two sites with the Wenatchee with the most continuous time series of data (located in Nason Creek [red circles] and Peshastin Creek [blue triangles]). Note, the range of axes vary by month to better show variability between predictions and measurements.

Figure 7 Example of accuracy and precision for average sites.

Measured stream temperature, predicted values from selected GAM models, and residual error for the entire time series of single sites selected from the Wenatchee River training dataset (A, Site WC503432-000155, Peshastin Creek) and testing dataset (B, Site #219, Nason Creek). These sites were chosen for display due to having near median values of RMSE out of all sites in the respective datasets and largely continuous timeseries. Horizonal dashed lines presented at −2 and 2 °C to provide guidance on size of residuals.

Table 5 Sensitivity analysis for the number of sites and years in fitting datasets.

Table showing the error of testing dataset predictions (RMSE Test) in model sensitivity analyses varying the number of years (A) and number of sites (B) utilized in the training datasets independently. RMSE Test was calculated from 100 iterations of randomly chosen sites/years out of all available for each sensitivity scenario (or all combinations if there were fewer than 100). The percent increase in prediction error from models fit with all data are shown in parenthesis. Model selection was not repeated for each scenario, and thus the same model formulas for each watershed was used for all scenario iterations. Note, sites have varying temporal coverage out of the entire time series of the fitting dataset and thus the removal of different sites/years represents different quantities of data. Percentage given for each watershed represents the average effective coverage percentage for each site in the fitting datasets (data days/[sites*years*365]).

 	Wenatchee (52%)	Chiwawa (56%)	M.F. John Day (33%)	Tucannon (67%)	
 	GAM	Linear	GAM	Linear	GAM	Linear	GAM	Linear	
A. Years	
6 (All)	1.26	1.37	0.85	0.85	1.55	1.80	0.95	0.98	
5	1.28 (2%)	1.38 (1%)	0.88 (3%)	0.85 (1%)	1.68 (8%)	1.88 (4%)	0.96 (1%)	0.99 (1%)	
4	2.13 (70%)	1.40 (2%)	1.33 (55%)	0.88 (4%)	1.94 (10%)	1.97 (10%)	0.99 (4%)	1.01 (4%)	
3	2.53 (101%)	1.68 (22%)	2.71 (217%)	1.03 (22%)	2.33 (51%)	2.07 (15%)	1.23 (30%)	1.09 (11%)	
B. Sites	
All	1.26	1.37	 	 	1.55	1.80	0.95	0.98	
35	1.32 (5%)	1.39 (1%)	 	 	1.68 (9%)	1.83 (2%)	0.95 (0%)	0.98 (0%)	
30	1.42 (14%)	1.41 (3%)	 	 	1.79 (16%)	1.86 (3%)	0.95 (0%)	0.98 (0%)	
25	1.63 (30%)	1.45 (6%)	 	 	1.94 (25%)	1.88 (5%)	0.96 (2%)	1.00 (2%)	
20	1.88 (50%)	1.54 (12%)	 	 	2.48 (60%)	1.97 (10%)	0.97 (2%)	1.01 (3%)	
15	2.58 (106%)	1.72 (25%)	 	 	3.14 (103%)	2.11 (17%)	1.06 (12%)	1.07 (9%)	
10	6.30 (401%)	2.81 (104%)	 	 	9.47 (511%)	3.07 (70%)	1.33 (40%)	1.26 (30%)	

The consistency in form and importance of the climatic relationships across the study watersheds suggests the presented methodology for parameterizing the effect of climate may be widely applicable. While the models as fit are only applicable to their respective basins, we saw largely consistent form in the modeled effects of climatic variables across all study watersheds. Three commonly cited challenges of modeling the influence of climate on stream temperature include hysteresis, temporal autocorrelation, and the temporally lagged influence of air temperature (Letcher et al., 2016). These issues are often described as independent processes and are addressed with distinct modeling techniques (e.g., distinct spring/fall models, temporal autocorrelation, air temperature time lags). However, each of these challenges is largely a consequence of the difficulty in accounting for spatial and temporal variation in the thermal inertia of streams which affects the rate at which water temperatures respond to influences such as heat transfer with the surrounding environment and solar radiation (Caissie, 2006). Thermal inertia varies spatially with channel form and stream size (Stefan & Preud’homme, 1993) and temporally with discharge levels (Smith & Lavis, 1975; Webb, Clack & Walling, 2003). Similarly, the influence of snowpack melt, which depresses stream temperatures, also varies spatially and temporally, both seasonally and interannually (Lisi et al., 2015). The ability of models to capture the dynamic nature of thermal inertia and the influence of snowpack with static variables or autocorrelation techniques which don’t vary across time and space is likely to be limited. However, while there is still room for improvement, our results suggest that including parameters incorporating the separate effects of discharge, snowpack, and air temperature and interaction terms accounting for the interdependency of these variables can be largely successful in this challenge.

Other spatiotemporal statistical modeling studies of stream temperature that depend on hierarchical and autocorrelation modeling techniques tend to see large increases in error when temporally expanding predictions beyond the range of the fitting data. This suggests that a major driver of stream temperature is not parameterized in the covariates. For example, Hocking, Neil & Letcher (2018) used a number of variables, interactions, random effects and an AR1 process to model stream temperature in the northeastern United States and saw an increase in RMSE from 0.59 °C for the training dataset to 2.06 °C when predicting the same sites but distinct years. While their model did attempt to account for variation in discharge and air temperature, it did not account for snowpack and depended heavily on a static AR1 process (AR1 = 0.77). We saw a proportionally much smaller decline in precision when predicting distinct years (average RMSE Train to average RMSE Test, 1.02 to 1.20 °C). This demonstrates the value of a full exploration of variables and interactions to account for spatiotemporal mechanisms.

Remaining patterns in our residuals suggest that our model fits could still be improved, potentially by accounting for autocorrelation. While well parameterized variables may be able to capture many of the physical forces that cause spatial and temporal autocorrelation in stream temperature, such as water volume/flow rates and the influence of snowpack, statistically parameterized variables are never likely to be perfect at this task given the complexity of climate/spatial influences that determine spatiotemporal patterns. However, all models would be improved by explanatory variables that better capture the dynamic influences on stream temperature. Numerous past studies have compared non-spatial models to spatial modeling techniques with non-spatial models generally performing poorly (e.g., Isaak et al., 2010; Turschwell et al., 2016). This has led to a general conclusion that non-spatial models are limited in their ability to predict complex patterns in stream temperature, particularly on shorter time scales such as daily measurements (Benyahya et al., 2007). However, our results demonstrate that even simple linear models are capable of producing relatively precise predictions assuming that a more comprehensive and effective list of covariates and interactions are parameterized.

Models generally described the majority of variability in stream temperature during warmer weather months (generally April-October) but were less useful in colder weather months. Due to the dependence of models on the influence of air temperature, models may be less able to describe variability as the relationship between air and stream temperature flattens as air temperatures approach freezing (Mohseni, Stefan & Erickson, 1998). Since the study basins experienced extended periods of time at below freezing air temperatures during the winter, stream temperatures had little variation across basins during these cold-weather months. If the winter is not of interest, the development of models could be restricted to exclude the winter period, as done by Letcher et al. (2016). However, due to higher variability in temperature during the summer in comparison to the winter (Fig. S4), winter-specific relationships may have also been overshadowed by summer variability. If winter temperatures are of concern, generating winter-specific models would potentially improve temperature predictions for this period.

GAM models generally outperformed linear models, particularly in the larger basins of the Wenatchee and the M.F. John Day. This is likely due to their ability to easily capture non-linear relationships, such as between air temperature and stream temperature (Holthuijzen, 2017), which may be more useful in more complex watersheds. Larger watersheds contain a wider variation in explanatory variable values and thus are more likely to include variable values that fall in the non-linear range of variable effects. Due to their better performance, we suggest utilizing GAM models over linear models, though we reiterate the need to highly restrict knots to prevent overfitting. Additionally, GAM relationships should be visualized to ensure that they align with hypothesized effects. Given a time series of four or more years, the ability of models to capture factors affecting interannual variability could be tested with a leave-one-year-out cross-validation procedure.

Models fit to smaller watersheds, such as the Chiwawa, or simpler networks, such as in the Tucannon, produced more accurate predictions and sensitivity analysis suggested that these watersheds required fewer sites. This result is not surprising as larger watersheds generally encompass higher variation in river/landscape characteristics which create challenges to modeling. Fewer sites per watershed than suggested by the sensitivity analysis would likely be required if training datasets contained fewer gaps and if logger locations were chosen to cover spatial gradients (Marsha et al., 2018). Sensitivity analyses suggested that at least four years of data was sufficient; however, including longer time-series and more diverse climate years would increase confidence in the ability of models to capture the influence of interannual climatic variability.

The accuracy of temperature predictions increased markedly when aggregating from daily values to mean monthly values, suggesting that the accuracy of the daily predictions can be considered a minimum for aggregated predictions. This result is not surprising as stream temperature has long been shown to be more directly correlated with air temperature at larger time scales (e.g., weekly and monthly) compared to daily values (Pilgrim, Fang & Stefan, 1998). Aggregating smooths out daily error caused by imperfect parameterization of thermal inertia and climate effects leading to mistimed or under/over predicted changes in temperature.

Model utility and potential improvements

Continuous predictions of stream temperature allow biologist and watershed managers to examine the effects of stream temperature throughout the entire year and to create temporally and spatially tailored summary metrics to a life stage or period of interest (e.g., mean temperatures during egg incubation or smolting). The level of accuracy of model predictions and the demonstrated ability of models to account for interannual variability in climatic influences is likely to be useful to biologists and watershed managers. As described above, spatially and temporally aggregating daily predictions to a period of interest is likely to improve accuracy and precision. The ability to expand predictions temporally and spatially suggests that such statistical models could be utilized to extend time series or fill-in monitoring gaps following the collection of enough data for parameterization, potentially reducing the requirements for effective long-term monitoring.

The ability of the models to successfully predict diverse climate years suggests that this approach may be effective for assessing the consequences of climate change. Climate change will cause increases in air temperature as well as significant reductions in snowpack, leading to earlier spring runoff and lower summer flows in the Pacific Northwest (Wu et al., 2012; Tohver, Hamlet & Lee, 2014). We demonstrate substantial impacts of air temperature, discharge, and snowpack as well as of the interactions between these variables and spatial effects on stream temperature. Consequently, models that do not account for all three of these variables and associated interactions are not likely to produce accurate climate change predictions. Estimates of changes in snowpack (e.g., Lute, Abatzoglou & Hegewisch, 2015), stream discharge (e.g., Chegwidden et al., 2017), and air temperature (e.g., River Management Joint Operating Committee RMJOC-II, 2018) as a consequence of climate change are widely available for the inclusion in future modeling efforts.

While results are encouraging, there are a number of ways that models could potentially be improved. As described above, the addition of autocorrelation methods to account for remaining patterns in residuals should be explored. The inclusion of autocorrelation techniques may allow for the use of statistical model selection (e.g., AIC), which we found to be overly permissive in our variable selection process. Models that don’t fully account for the interrelatedness of data points, either through variables or autocorrelation techniques, may demonstrate inflated variable significance (Isaak et al., 2014). Improvements may also come from continued advancements in variable parameterization, for example, the inclusion of spatial variables that more thoroughly account for local stream characteristics impacting stream temperature (Holtby, 1988; Johnson, 2004; Caissie, 2006), such as reach-specific shading, solar radiation, and groundwater influences (McNyset, Volk & Jordan, 2015; Turschwell et al., 2016; Isaak et al., 2017b). These localized factors have a substantial influence on stream temperature (Holtby, 1988; Johnson, 2004; Caissie, 2006) which can occur within short reaches (Johnson, 2004). As currently constructed, models are basin specific as environmental relationships are fit relative to measurements at the specific environmental monitoring stations. Incorporating spatially explicit estimates of climate variables, instead of data from point sources, would potentially improve general applicability of models since environmental values would be specific to each stream segment (e.g., DayMet in Hocking, Neil & Letcher, 2018). Spatially explicit climate variables would also reduce the need to include interactions to describe variation in climate effects across space, thereby allowing for simpler models and interpretation.

Conclusions

The determinants of spatiotemporal patterns in stream temperature are complex, driven by seasonal and interannual climate variability interacting with diverse landscapes. Our results suggest that the inclusion of variable interactions in statistical models based on mechanistic principles can produce accurate stream temperature predictions across space and time. We demonstrate that this can be achieved with simple modeling techniques informed by easy to parameterize variables developed from widely available environmental and geographical information. These methods allow for the filling of temporal gaps in stream temperature monitoring records and the ability to predict unmonitored years. However, a level of remaining patterns in residuals suggest that this methodology could still be improved, either through the inclusion of methods to account for autocorrelation or the parameterization of missing variable influences. We have suggested a number of potential improvements that we believe will increase the precision of predictions, reduce the need for separate seasonal fall and spring models, and make models more generally applicable across larger spatial regions and levels of climate variability.

Supplemental Information

Supplemental Information 1 Interpretation and visualization of variables and interactions

This document provides a more in-depth interpretation and visualization of the variable effects and interactions included in the stream temperature models than could be provided within the article.

Click here for additional data file.

Figure S1 Air temperature comparison between training and testing dataset years for the Wenatchee

Daily mean air temperature measurements for the Wenatchee for all years in the fitting dataset (red) and validation dataset (blue). Smoothed GAM fit for all years of the fitting and validations datasets are shown by thick red and blue lines respectively.

Click here for additional data file.

Figure S2 Snow depth comparison between training and testing dataset years for the Wenatchee

Daily mean snow depth measurements for the Wenatchee for all years in the fitting dataset (red) and validation dataset (blue). Smoothed GAM fit for all years of the fitting and validations datasets are shown by thick red and blue lines respectively.

Click here for additional data file.

Figure S3 Discharge comparison between training and testing dataset years for the Wenatchee

Daily mean discharge measurements for the Wenatchee for all years in the fitting dataset (red) and validation dataset (blue). Smoothed GAM fit for all years of the fitting and validations datasets are shown by thick red and blue lines respectively.

Click here for additional data file.

Figure S4 Example of process for splitting spring and fall seasons for modeling

All stream temperature in the Wenatchee fitting dataset shown by day of year with GAM smoother shown by red line and date of predicted max temperatures marked by blue line. Dates of predicted max temperature in fitting datasets were utilized to split year into spring warming and fall cooling periods for model fitting and subsequent validation predictions.

Click here for additional data file.

Figure S5 Example of over-fit GAM surface and the smoothing effect of limiting splines

Example of GAM models fit utilizing Wenatchee River fitting dataset data with effects of averaged air temperature the five days before the predicted day (T5_a) and the average catchment area elevation (AE) utilizing penalized regression splines to determine the number of knots (a) and with knots specified at 3 (b). Using penalized regression splines produces a “crumpled blanket effect”, which is overfit and does not align consistently with hypothesized effects of the variables.

Click here for additional data file.

Figure S6 Map showing spatial regions in the Wenatchee watershed and CHAMP temperature logger sites

Model training dataset prediction biases from selected GAM models are shown for these spatial regions in Fig. S7.

Click here for additional data file.

Figure S7 Accounting for spatial variability in the selected Wenatchee models

Boxplots showing the residuals by spatial regions (A and C) for the Wenatchee selected GAM spring (A) and fall (C) models, as well as for all residuals combined (B and D) in red compared against analogous predictions from a smoother for day of year fit to the entire basin in turquoise (smoother shown in Fig. S4). Selected models reduce spatial biases. Spatial regions analyzed are shown in Fig. S6 (ChR is Chiwawa River, LWR is Little Wenatchee River, M/UP is Mission and Upper Peshastin Creeks, NC is Nason Creek, SEH is Southeast Hills, SWM is South West Mountains, MS is Mainstem, WR is White River)

Click here for additional data file.

We thank Kris McNyset for performing much of the foundational work that this project was based on. Thanks to Arielle Gervasi and Jesse Langdon for GIS processing. Thanks to Chris Jordan for a friendly review. Finally, thanks to Aimee Fullerton for productive discussions, feedback and insights.

Additional Information and Declarations

Competing Interests

Author Contributions

Data Availability

Jared Siegel is currently employed by Ocean Associates, Inc., contracted to the National Marine Fisheries Service. He was formally employed by South Fork Research, Inc. Carol Volk is the owner of South Fork Research Inc. and is also currently employed by the City of Seattle.

Jared E. Siegel conceived and designed the experiments, performed the experiments, analyzed the data, contributed reagents/materials/analysis tools, prepared figures and/or tables, authored or reviewed drafts of the paper, approved the final draft.

Carol J. Volk conceived and designed the experiments, contributed reagents/materials/analysis tools, authored or reviewed drafts of the paper, approved the final draft.

The following information was supplied regarding data availability:

Data and R code for this project are available on the GitHub page of South Fork Research: https://github.com/SouthForkResearch/TemperatureModels_Predictive.

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
