# Peer review of "Accurate spatiotemporal predictions of daily stream temperature from statistical models accounting for interactions between climate and landscape"

_PeerJ, doi:10.7717/peerj.7892_

## Round 0.1 · original submission · Major Revisions

Both reviewers provided positive and constructive comments. The primary concern that requires attention was raised by Reviewer 1, who questioned the lack of consideration of spatial and/or temporal autocorrelation in the statistical models used. That reviewer provided substantial background information and very thorough supporting discussion which will require significant attention in the revised manuscript. Please provide a specific response to each comment provided by the reviewers and note where and how you have modified the manuscript accordingly.

Reviewer 1 ·

Basic reporting

This paper develops models of daily mean river temperature using both linear and non-linear (gam) models with interaction terms, for 4 sub-catchments in the Columbia River. The performance of these models, in each of the catchments they were developed for, are then compared using RMSE for the training dataset, leave-one site out cross validation for training dataset and prediction to a testing dataset. The paper also explores the optimal number of sites and years required within the model and the accuracy of predicting summary metrics (e.g. monthly values) from the predicted daily river temperatures.

Generally, the paper is well written and detailed; in particular the introduction provides a nice overview. A broad range of literature is also cited. There are some interesting covariates explored, with associated reasoning. The figures and tables are well produced and the supplementary information extensive. The data sources are also provided within the methods, although some data does not appear to be available for public download currently.

I think the paper would benefit from a clearer explanation of the originality/novelty, given the number of published methods comparisons and models of river temperature. If reductions in the papers’ length were required, I would suggest that the exploration of aggregating to monthly metrics and the sensitivity analysis could be removed. I do not feel strongly about this though.

However, I do have some major concerns about the methods and specifically the statistical approach undertaken, which I will outline in the ‘experimental design’ box. I have also provided further specific comments, minor questions and references relating to this review in the ‘general comments for authors’ box.

I hope you find this review to be useful and constructive.

Experimental design

The methods section is detailed but does require some further clarification in areas:

1. My main concern regards the validity of the statistical approach. Specifically, I do not think that using interaction terms allows temporal or spatial autocorrelation to be accounted for. The assumptions of independence, required for these types of model, have still been violated. Not accounting for autocorrelation could result in covariates being retained within overly complex (over-fitted) models where there is not real statistical significance for their inclusion, as tests of statistical significance are too generous meaning type 1 errors occur more often (Isaak et al., 2014). For example, similarly to this study, a non-spatial river temperature model produced by Isaak et al., (2014) contained all significant predictor variables. However, when the same covariates were included in a spatial model only 2 out of the 5 were significant. As such, models which do not account for autocorrelation may not truly be suitable, responses could seem reasonable but may still be spurious and drawing inferences could be problematic. It is likely that this would also then have consequences for making predictions to new locations/time periods.

Residual analysis is often undertaken to establish if a model can be accepted (e.g. Laanaya et al., 2017) and if conclusions can be drawn from it (such a process is demonstrated nicely in this ‘fromthebottomoftheheap’ blog post: www.fromthebottomoftheheap.net/2014/05/09/modelling-seasonal-data-with-gam). It was good to see that temporal patterns remaining in the residuals were honestly highlighted (Figure 7) and discussed as a limitation (lines 636-637). In addition, it would be interesting to see if there were also spatial patterns remaining in the residuals (e.g. semivariograms or torgegrams), given the proximity of the sites to each other (Figure 1). Please could further details be provided to explain why the models were still accepted even with patterns in the residuals, which suggest some unsuitability?

2. Please provide further justification for why only linear and GAM models are compared. One of the objectives is to illustrate how variability commonly accounted for by hierarchical or autocorrelation statistical modelling techniques could instead be parameterised by interactions. Consequently, I would have expected comparison to a model which accounts for spatial and temporal autocorrelation. Although previous papers have shown that models which account for spatial and temporal autocorrelation perform better than models which don’t, while the magnitude of this difference varies depending on the extent of the structure in the residuals (Isaak et al., 2014).

3. Four models were produced for each catchment; linear spring, linear autumn, gam spring and gam autumn. Given that one of the benefits of gam models is being able to fit smooth terms (e.g. cubic cyclic spline for day of the year) it is unclear why the models were split into spring and autumn. Was this only to enable comparison to the linear models, where the year would have to be split to allow a linear day of the year covariate? How were the numbers of knots chosen for smooth terms and which covariates were the linear interaction terms? Further information on the model selection procedure would also be useful. Currently, it is unclear exactly how model selection was undertaken. For example were all possible model combinations compared or was it step-up step-down model selection from a full starting model? Additionally, use of an information criterion (e.g. AIC, BIC) to compare models would account for goodness of fit and the number of model parameters. This would also ensure complete comparability as linear and gam models would be ranked using the same metric (rather than R2 for linear models and deviance explained for gam). The performance of the fit of the final models (R2/deviance explained), does not appear to be provided. This would be useful and could potentially be added as rows in Table 3.

4. Inferences about the potential wider use of this approach are made. However, currently all of the model performance metrics are undertaken for the catchment that the model was developed in (so models are catchment specific) and with both the training and testing datasets covering similar spatial extents but differing time periods. It would have been interesting to explore how well the models predict to unobserved locations in distinct areas of the catchments (e.g. lower Middle Fork John Day, tributaries of the Tucannon and western tributaries of Wenatchee), particularly as issues with spatial outliers were also observed (Line 459). Wider applicability/performance could potentially have been tested for explicitly by either fitting a full model to all catchments or by testing model performance between catchments by taking a catchment model and predicting in a different catchment (e.g. Jackson et al., 2017, 2018).

5. Response shapes are categorised as ‘universal’ and ‘local’ to infer wider applicability of the models but this is not explicitly shown. A short explanation of the final models is provided, with Figure 3 illustrating an example interaction. The majority of the discussion on the model relationships is contained within the supplementary material but plots are not provided for all catchment models to allow comparison. It appears as though, for each interaction, 1 of the 4 catchments is shown for illustrative purposes. A multi-panel plot of each individual covariate interaction in each catchment and season may be useful for demonstrating consistency in response shape and effect size.

6. It is good to see that the raw data sources are provided in the methods. I would like to see further information on why the training and testing datasets were collected from two different databases (Columbia Habitat Monitoring Programme and NorWest Regional Stream Temperature Database), rather than the testing dataset being a random sample of all of the sites/years available. For example, was this to enable testing predictions in a distinctly different time period and if so why? Although, quality control is assumed to have been undertaken by the data providers, with further visual checks undertaken prior to analysis, it could be possible that quality control and monitoring procedures (sensor/logger types and calibrations) vary between the two databases. This could introduce systematic differences between the training and testing datasets, that is an artefact of data collection rather than varying river temperatures, which may influence predictive performance. Furthermore, Table 1 provides information on the data available but what was the minimum number of data points required before a site could be used? Alternatively, were all sites used regardless of dataset length, as daily values were of interest and not summary metrics? Given the objective to model complex spatial and temporal variability in river temperature, clarification on why some sites with distinct (potentially groundwater driven) regimes were removed would also be useful. Were these sites actually monitoring temperatures in groundwater springs rather than a section of river influenced by groundwater inputs?

7. Table 2 provides a clear and concise description of the potential spatial and temporal model covariates. Temporal covariates (e.g. air temperature, discharge) were taken from single points in each catchment - was this purely down to data availability? Please provide justification of this choice. Specifically, because using the closest monitoring station or station with the most similar characteristics (if multiple stations were available) to the river temperature site may have also provided some explanation of spatial variability in climatic controls (e.g. Ta gradients/lapse rates) rather than needing to use the landscape proxies such as catchment elevation.

Validity of the findings

I am uncertain about the validity of the findings given my concerns on the methodology (details above). In brief, I have concerns that some of the model covariates may be included where a true ‘significant’ relationship is absent due to the non-independence of daily river temperature data in space and time. This is despite the exploration of the physical plausibility of the relationships in the models provided in an extensive supplementary document. It is good to see the patterns remaining in the residuals honestly described as a limitation (and illustrated in Figure 7). However, I would still argue that, as a result of assuming that observations are all independent when they are not, the models could be over-fitted, significant covariates could be spurious and parameter estimates could be driven by spatially clustered sites in the network (Isaak et al., 2014). As such, it may be inappropriate to accept and draw inferences from these models. Setting these issues aside, no models which account for autocorrelation are fitted to allow an explicit assessment into which approach performs best or to assess if the interactions would still be included as significant covariates in a model where non-independence is accounted for. Thus any suggestions that these models would outperform or have more utility than a model accounting for autocorrelation would be speculation.

There are some inferences around the wider applicability of the findings but currently each model is catchment specific and the performance metrics refer to performance at new sites and time periods within that catchment. Furthermore it seems that outlying climatic years were also removed from the testing dataset (lines 265-289), which could potentially improve model performance by ensuring predictions were not beyond the observations within the fitting period. The majority of the covariates and interactions included in the final model are contained within the supplementary material but plots do not seem to be provided for all catchment models to allow comparison of response shapes, effect sizes and importance in the models.

Information on the datasets used is stated in the methods. However, model training and testing datasets also came from two separate regional databases, potentially introducing differences as a result of different sensor/logger types and calibration procedures. This should also be acknowledged as a potential issue.

Additional comments

Here I have included specific comments for your consideration and more minor queries.

Line 82 – also difficult to upscale deterministic models

Line 88-89 – not necessarily easier to implement as they come with their own set of technical challenges and data requirements, as described by Letcher et al., (2016).

Lines 114 – please delete ‘Ashley’ for consistency with other references

Line 117 – applicable over the area they were fit for e.g. sites, catchments or regions?

Line 119-123 – given the references provided, I am not sure how the models are ‘limited in either their spatial or temporal flexibility’. The model developed by Li et al., (2014) covered 5 US states and could predict for every day of the year. The model developed by Segura et al., (2015) covered the conterminous USA and could predict monthly and seasonal river temperatures. The model developed by Jackson et al., (2018) covered Scotland and could predict for every day of the year.

Line 130 – although there are some examples where spatially and temporally (daily) continuous estimates of stream temperature could be produced (e.g. Li et al., 2014; Piccolroaz et al., 2016; Sohrabi et al., 2017; Jackson et al., 2018)

Lines 135-138 – However, by accounting for spatial and temporal correlation you would have a better model fit where all covariates are significant, rather than potentially spurious. For example, Isaak et al., (2014) provide a good overview of why autocorrelation is important and state that where autocorrelation exists, and is not accounted for, you could encounter the following problems:
1. ‘Database contains less information than the number of measurements implies’
2. ‘Tests of statistical significance are too liberal and type 1 errors occur more frequently’
3. ‘Parameter estimates may be biased or driven by spatially clustered sites’

Line 194 – why was the Chiwawa River separated out of the Wenatchee River basin?

Line 245-248 – am I correct in thinking that this was because spatially continuous predictions of climatic variables (e.g. gridded air temperature datasets) were not available? If so, a sentence stating this would be useful.

Line 274-277 – this identifies broad spatial patterns across the catchment but does not account for non-independence of sites close together. For example, Isaak et al., (2014) showed upstream catchment area to have a p value of 0.01 in a non-spatial model and 0.57 in a spatial model accounting for spatial covariance, suggesting a false detection of an effect in the non-spatial model.

Lines 284-297 – it seems like many of these interactions are used because there is not a spatially (and temporally) continuous air temperature or discharge dataset, is this correct?

Lines 510-514 – this is arguably conjecture as the ability to predict to unmonitored catchment locations (e.g. lower Middle Fork John Day, tributaries of the Tucannon and western tributaries of Wenatchee) and out with the catchments that the models were developed in (e.g. Tucannon predicting in Wenatchee) has not been tested.

Lines 527-529 – consistency in form and importance of climatic relationships across catchments does not seem to be shown. Have I missed this somewhere?

Lines 542-548 – comparing the performance of these models to models which account for spatial and temporal autocorrelation would be valuable in this respect.

Lines 558-560 – from the information provided in lines 265-268 it seems like the test dataset was compared to the training dataset to ensure that there were not outlying climatic conditions which would reduce prediction confidence. Is this correct? If so, could this be in part explaining why the models appear to have a smaller decline in precision when predicting to new years of data?

Lines 577-579 – I am not sure what is meant by this, given that the two model types in this study used the same input datasets? Please clarify.

Lines 593-595 – could this also be to do with the heterogeneity in river characteristics observed in these smaller catchments rather than inherently being to do with the size or network characteristics?

Lines 636-648 – even with improved covariate characterisation I still think the issues around spatial and temporal autocorrelation (non-independence of data) would need to be addressed.

Table 1 – Please define what you mean by effective coverage. Am I correct thinking that ‘data days by month’ is across the whole time period? A note clarifying this would be useful. I wonder if rather than ‘data days’ number of sites with days above a set threshold of days (e.g. 95%) would be more intuitive for understanding data availability.

Table 2 – Table 2 is very useful. Adding in ‘of the site’ at the end of the description in the ‘Spatial’ column for those metrics which are generated for each river temperature sites’ upstream catchment area (rather than a single whole catchment value) would further improve clarity (e.g. ‘Area’, ‘Lake’ and potentially ‘BFI’).

Table 3 – Am I correct that the grey cells in the table denote covariates included in the final model? If so, a note in the table caption would be useful to avoid potential confusion. I also wonder if something about the covariate importance in the model would be useful.

Supplementary document – this document is extensive and provides suggestions as to the physical process reasoning behind the significant interactions, which is covered to a limited extent in the manuscript itself.

Supplementary figures – these figures give an overview of the climate conditions in both training and testing datasets.

References:
Isaak, D.J., Peterson, E.E., Ver Hoef, J.M., Wenger, S.J., Falke, J.A., Torgersen, C.E., Sowder, C., Steel, A.E., Fortin, M.-J., Jordan, C.E., Ruesch, A.S., Som, N., Monestiez, P., 2014. Applications of spatial statistical network models to stream data. WIREs Water 1, 227–294.

Jackson, F. L., Fryer, R. J., Hannah, D. M., Millar, C.P., and Malcolm, I. A. (2018) A spatio-temporal statistical model of maximum daily river temperatures to inform the management of Scotland's Atlantic salmon rivers under climate change. Science of the Total Environment. 612, 1543-1558.

Jackson, F.L., Fryer, R.J., Hannah, D.M., Malcolm, I.A.,(2017) Can river temperature models be transferred between catchments? Hydrology and Earth Systems Science. 21, 4727-4745

Laanaya,F, St-Hilaire A, Gloaguen E., (2017) Water temperature modelling: comparison between the generalized additive model, logistic, residuals regression and linear regression models, Hydrological Sciences Journal, 62:7, 1078-1093.

Letcher, B.H., Hocking, D.J., O’Neill, K., Whiteley, A.R., Nislow, K.H., O’Donnell, M.J., (2016). A robust hierarchical model of daily stream temperature using air-water temperature synchronization, autocorrelation, and time lags. PeerJ 4:e1727, 3:e1971.

Li, H., Deng, X., Kim, D., Smith, E.P., (2014). Modelling Maximum Daily Temperature Using a Varying Coefficient Regression Model. Water Resources Research, 50, 3073-3087

Piccolroaz, S., Calamita, E., Majone, B., Gallice, A., Siviglia, A., and Toffolon, M. (2016) Prediction of river water temperature: a comparison between a new family of hybrid models and statistical approaches. Hydrol. Process., 30: 3901–3917.

Segura, C., Caldwell, P., Sun, G., McNulty, S., Zhang, Y., 2015. A Model to Predict Stream Water Temperature across the Conterminous USA. Hydrol. Process. 29, 2178–2195.

Sohrabi, M.M., Benjankar, R., Tonina, D., Wenger, S.J., Isaak, D.J., 2017. Estimation of Daily Stream Water Temperatures with a Bayesian Regression Approach. Hydrol. Process.

·

Basic reporting

I thought this paper was well-organized and very well done. Most of my comments pertain to points that I believe need further clarification.
The Introduction is good. The authors set up the problem and the needs well. The Methods are clear and statistically sound but I have identified some points that could be clarified. The Results and Discussion sections are comprehensive, are well-supported, and address the needs identified in the Introduction. There is a discussion of future research needs going forward as well. I did not find much to criticize with this manuscript.
I didn’t see the raw data so I wasn’t able to review or evaluate that. But the Supplemental Information explaining the physical basis for the interaction terms was very informative, probably more so than seeing the raw data.

Experimental design

The research questions were well defined. I had questions about the Methods that I think should be addressed. These are listed in the General Comments to the Author.

Validity of the findings

No comment

Additional comments

Line 157-160. The authors are presenting two different and unique approaches to stream temperature statistical modeling – the incorporation of interaction terms in the model and the use of GAMs. In these lines, they discuss the pros and cons of GAM models. They should come back to this point in the Results and Discussion. Was there any tendency for the GAMs to overfit data, as suggested here? What were the benefits and is it recommended that these be used instead of simple linear models with interaction terms? This was addressed in the Discussion but I didn’t come away with a strong sense that I had answered these questions.
Line 221 – watershed should be plural.
Line 262-263 and lines 392-393. I’m having a little difficulty understanding how snowpack depth and stream discharge do not give you the same information and therefore result in issues of multicollinearity? After reading the Supplemental Information, I think this is clarified in that material but I would include it in the main paper too. Furthermore, why do you need Apr 1st snowpack depth (Table 3) when you have snow depth for all days? This should be explained more clearly in the paragraph from line 390-412, where you discuss the use of both Snow Depth and Apr 1st Snow Depth.
Line 308-309. How were the spring and fall season separated on the other end of the year (in winter) with fall going into spring? Ice formation? Calendar Year?
Line 329 – is this supposed to refer to Table 1? I don’t see the relevancy of Table 1 to hypothesized effects. Is this the correct reference?
Lines 417-420. Here again, I’m having trouble understanding why the information on high tributaries versus low tributaries wasn’t captured with the Discharge term. I believe this is because you didn’t have discharge for the tribs, only the downstream end of the mainstem, but I would make this clear.
Line 460 – did you mean “better or comparable precision…”
Line 499-501 - Sensitivity analysis of years – 4+ years is very good. How does this compare to other results?
Line 587 – Please indicate how one can ensure that GAM models are not over-fitting since this is emphasized a couple of times as the negative side of GAMs – is this done through a testing dataset process or some other means?
Line 587 – “…model relationships…” or …the modeled relationship…”
Line 592 – The authors should explain why capturing a non-linear relationship is more useful in larger, more complex watersheds.
Table 2 – the descriptions for the temporal column for variables T3a and T5a appear to be reversed.
Table 2 – I’m not really understanding the variable Snow Depth and how it is different from Apr 1st snow depth. Is it collected on a particular date or was it a continuously collected throughout the season? Wouldn’t values toward the end of the season (once it starts to melt) be more important for stream temperature anyway? And why do you need Apr 1st snow depth if you have snow depth from the entire season?

---

## Round 0.2 · accepted · Accept

This paper will make a nice addition to the literature on modeling the spatio-temporal dynamics of stream temperature, which is an important concern in a warming climate.

·

Basic reporting

No comment

Experimental design

No comment

Validity of the findings

No comment

Additional comments

I just noticed a few errors that need correction. I did not review the article extensively for corrections though.

Line 160 – “non-linear” should be “non-linearity” or “non-linearities”
Line 375 – correct “Since we contend that the RMSE test. represents the truest test of the ability of models”
Line 396 – “The mean RMSE test. for all iterations for each site count”
There are several other sentences with “RMSE test.” that need correcting. Search for “RMSE test.” and correct the punctuation where needed.